# Fluorescence calibration method for single-particle aerosol fluorescence instruments

Ellis Shipley Robinson<sup>1,2,†</sup>, Ru Shan Gao<sup>1</sup>, Joshua P. Schwarz<sup>1</sup>, David W. Fahey<sup>1</sup>, and Anne E. Perring<sup>1,2</sup>

<sup>1</sup>NOAA Earth System Research Laboratory, Boulder, CO, USA

<sup>2</sup>Coorperative Institute for Research in Environmental Sciences, Boulder, CO, USA <sup>†</sup>Now at Center for Atmospheric Particle Studies, Carnegie Mellon University

*Correspondence to:* A.E. Perring (anne.perring@noaa.gov)

**Abstract.** Real-time, single particle fluorescence instruments used to detect atmospheric bioaerosol particles are increasingly common, yet no standard fluorescence calibration method exists for this technique. This limits the utility of these instruments as quantitative tools and complicates comparisons between different measurement campaigns. To address this need we have developed a

- method to produce size-selected particles with a known mass of fluorophore, which we use to calibrate the fluorescence detection of a Wide-band Integrated Bioaerosol Sensor (WIBS-4A). We use mixed tryptophan-ammonium sulfate particles to calibrate one detector (FL1; excitation = 280nm; emission = 310-400nm), and pure quinine particles to calibrate the other (FL2; excitation = 280nm; emission = 420-650 nm). This procedure allows users to set the detector gains to achieve a known
- absolute response, calculate the limits of detection for a given instrument, improve repeatability of instrumental set-up, and facilitate intercomparisons between different instruments. We recommend calibration of single-particle fluorescence instruments using these methods.

## 1 Introduction

Primary biological aerosol particles (PBAP) are of wide interest due to their potential impacts on
air quality (e.g. Prussin II et al. 2015), ecology (e.g. Morris et al. 2013), and Earth's climate (e.g. Creamean et al. 2013). PBAP comprise a broad class of atmospheric particles ranging from the small (viruses, as small as ~20 nm diameter) to the large (pollen grains, 5-100µm diameter) with bacteria, fungal and plant spores, and plant, insect, and animal fragments in between (Després et al., 2012). Despite the ubiquity of PBAP, many important questions remain regarding their atmospheric

impacts.

The measurement of atmospheric PBAP has historically involved off-line techniques, such as culture-based methods and manual cell-counting by optical fluorescence microscopy. These methods require long air sampling periods, significant post-collection labor, and provide poor temporal resolution. In response to these shortcomings, a new generation of on-line, automated instruments for the measurement of PBAP, such as aerosol mass spectrometers (Tobias et al., 2005) and fluores-

cent particle spectrometers (Pan et al., 2003; Kaye et al., 2005), have recently been developed.

Measurements of single particle fluorescence has been used for rapid detection of PBAP in the fields of atmospheric science (Pöschl et al., 2010), public health (Bhangar et al., 2015), and biological warfare research (Greenwood et al., 2009). Many biological compounds, including certain amino

acids (e.g. tryptophan, tyrosine), metabolic small molecules (e.g. the reduced form of nicotinamide adenine dinucleotide, or NADH), and some proteins (e.g. Green Fluorescent Protein), are intrinsically fluorescent (Chudakov et al., 2010). In single-particle fluorescence instruments, fluorescence in such compounds is induced using ultraviolet excitation energy, and the resulting fluorescence is detected either in relatively broad emission bands using filters or with spectral resolution using a spectrometer. Fluorescent particle loadings are then used as a proxy for PBAP.

Despite the proliferation of single-particle fluorescence instruments (see Pan et al. 2003 and Kaye et al. 2005 for early prototype examples, and the Ultra-Violet Aerosol Particle Sizer (UV-APS; TSI, inc.) and Wideband Integrated Bioaerosol Sensor (WIBS; DMT, Inc.) for commercially-available examples), there is no standard method used to calibrate the magnitude of their fluorescence

signals. Fluorescently-dyed polystyrene latex spheres (FPSLs) are commonly used to assess detector performance, instrument alignment and excitation pulse timing. FPSLs, however, have significant batch-to-batch variability and suffer from poor shelf-life; thus they do not provide a repeatable, absolute calibration for fluorescence intensity. The lack of a standard limits our ability to compare observations made with different instruments, to track long-term instrument stability, and to assess 45 the fundamental limit of detection of the technique.

The amount of fluorescently emitted light that is measured is also a potentially useful metric for fluorescent particle attribution in single-particle fluorescence instruments. The use of these fluorescence magnitudes, however, varies widely in the published literature. Several recent studies have employed a binary yes-no classification of fluorescence above a threshold (e.g. Gabey et al. 2010,

- Perring et al. 2015), essentially ignoring the fluorescence magnitude beyond the threshold. Fluorescence magnitudes have been used as input variables in automated particle-clustering analyses (Robinson et al., 2013; Crawford et al., 2015b) and to manually sort sampled particles into groupings (Wright et al., 2014). Particles emitting so much fluorescent light as to saturate fluorescence detectors are sometimes excluded from analysis (e.g. Toprak and Schnaiter 2013) and relatively
- weak fluorescence has been proposed as a possible discriminator of interfering non-biological particles (Hill et al., 1999; Crawford et al., 2014, 2015a; Yu et al., 2016). The utility of fluorescence magnitudes will increase greatly with the development of an absolute fluorescence calibration strat-

egy applicable to any single-particle fluorescence measurement technology.

Here we present a reliable calibration strategy for fluorescence intensities measured by single-

60 particle fluorescence instruments. Methods for solution preparation, particle generation, and data analysis are presented.

#### 2 Materials & Methods

We evaluated the response of a wideband integrated bioaerosol sensor (WIBS-4A; Droplet Measurement Technologies; Boulder, CO, USA) to monodisperse aerosol particles containing a known
mass of fluorescent material. These experiments were conducted using fluorophores emitting in one or more of each of the fluorescent detectors and for different detector gains in the WIBS-4A. The following criteria guided our selection of fluorescent material:

- Fluorescent properties: fluorophores were chosen to match one or more of the excitation wavelengths (280, 370 nm) and emission bands (310-400, 420-650 nm) of the WIBS-4A.
- Stability: chemically inert fluorophores were chosen such that the signal from particles of a given size were constant over the course of a calibration.
  - Repeatability: the relationship (calibration curve) between fluorescence signal and fluorophore mass needed to be repeatable across multiple experiments with different batches of prepared solutions.
- 4. Availability & ease of preparation: all fluorophores used are inexpensive and easy to acquire. Importantly, each fluorophore chosen was soluble either in water or isopropanol for atomization.
  - 5. Safety: the materials used are all relatively safe to handle and prepare, though proper personal protective equipment was worn and exposure to exhausted particles was avoided.
- Tryptophan and quinine fulfilled these requirements. NADH and naphthalene were also tested but each failed to meet one or more of the above requirements. Results from all materials tested are presented in Section 3. Below we present our detailed strategy for calibrating the fluorescence signals from the WIBS-4A with these materials.

# 2.1 Wideband Integrated Bioaerosol Sensor (WIBS-4A) operation

We validated the procedure using a commercially-available WIBS-4A, first described by Kaye et al. (2005) and later in significant detail by e.g. Gabey et al. (2010) and Perring et al. (2015). We will briefly describe its operating principles and the instrument settings used in this study.

The WIBS counts and sizes all incoming particles using elastic scattering from a continuouswave laser (635 nm, 12 mW). This scattering signal triggers the sequential flashing of two Xenon

- lamps (5W L9455 modules, Hamamatsu Photonics K.K., Japan), one of which is filtered to emit light at 280 and the other at 370 nm. Any resulting fluorescent light is collected by two photomultiplier tubes (PMTs, H10720-110, Hamamatsu Photonics K.K., Japan) filtered to detect only specific wave bands: the FL1 detector detects 310-400 nm emission, and the FL2 detector detects 420-650 nm emission, though the peak sensitivity for each detector is in the 350-450 nm range. A
- reference voltage input controls the gain on each PMT, which is changed manually with a variable potentiometer. We refer to this as the gain voltage throughout the rest of the paper. The FL2 detector also detects the scattering signal used for optical sizing. At reasonable particle sample rates, it can do so without interfering with the fluorescence measurement since the scattering event and the two flash lamp pulses are separated in time. Three fluorescence signals are therefore recorded for
- a given particle: fluorescence between 310 and 400 nm following 280 nm excitation (referred to here as Channel A) and fluorescence between 420 and 650 nm with either 280 or 370 nm excitation (referred to as Channels B and C, respectively).

Both before and after the fluorescence calibration, the WIBS was run in forced trigger (FT) mode. In FT mode, the two xenon flashlamps are triggered automatically (as opposed to being triggered

- by the presence of a particle) at ~2 Hz, to assess the background light detected in each PMT in the absence of particles. The FT background in each detector is a function of the flash lamp intensity, the flash lamp alignment, the efficiency of the filter at rejecting the excitation wavelength, the detector gain setting, and fluorescence from any material deposited within the instrument cavity (Toprak and Schnaiter, 2013). In general WIBS data analysis, FT data are used to determine a signal threshold
- for each channel above which a particle is considered fluorescent. In ambient measurements, where a majority of particles are non-fluorescent, the fluorescent threshold can be assessed without taking the instrument off-line to run in FT mode, as there generally exists a dominant population of nonfluorescent particles that have a distribution of fluorescence magnitudes identical to the background data collected in FT mode (Perring et al., 2015). Here, sample particles were fluorescent by design
- and we use 2-5 minutes of FT mode data to determine fluorescence thresholds for each channel. Gaussian functions were fit to FT signal peak intensity and fluorescence thresholds were defined as three standard deviations above the center of the fitted gaussian function (FT +  $3\sigma$ ).

All experiments presented here use a WIBS sample flow rate of 0.3 liter/min, and a sheath flow rate of 2.1 liter/min, close to typical factory settings. Laboratory tests reveal an inverse relation-

120 ship between optical size and particle velocity, as shown in Figure 1. This relationship likely is attributable to insufficient signal processing speed to fully resolve the scattering peak magnitude, a problem exacerbated at higher particle velocities. Due to this flow sensitivity, as well as flow dependence for the flash lamp timing, we note that instrument calibrations should be performed at the same flow rate that will be used in ambient or lab measurements.

#### 125 2.2 Particle generation and sampling

The experimental setup used for WIBS-4A calibration consisted of three general components: particle generation, particle conditioning, and measurement. This is shown schematically in Figure 2.

Fluorescent particles were generated by nebulizing a solution containing a fluorophore dissolved
in either isopropanol (99.9% purity, HPLC-grade, Pharmco) or deionized water, depending on the solubility of the fluorescent material. Additional non-fluorescent material was added to the nebulized solution, as needed, to adjust the per-particle mass of fluorophore. The desired range of fluorophore masses was determined empirically based on typical factory gain settings and previous observations of fluorescent magnitudes of known biological materials. For example, pure quinine (>98%, Sigma-

- Aldrich) particles within the size range of the WIBS (>0.8 um) exhibited fluorescence intensities that were within the dynamic range of the detector at typical gain settings. For tryptophan (L-tryptophan, >98%, Sigma-Aldrich) on the other hand, pure particles saturated the detector at typical gain settings and produced much higher fluoresence signals than biological materials of comparable size. That the instrument is more sensitive to tryptophan than quinine on a mass basis is likely due to
- the peak sensitivity of the WIBS-4A PMTs overlapping significantly with the tryptophan emission spectrum, and less so with that of quinine (Pant et al., 1990; Goldberg et al., 2012). Therefore tryptophan-based calibration particles were an internal mixture of tryptophan and ammonium sulfate "filler." Ammonium sulfate (≥99%, Sigma-Aldrich) was chosen because it is very soluble in water and has previously been shown not to fluoresce in the WIBS (Toprak and Schnaiter, 2013). In this
- case the nebulized solution was prepared by mixing appropriate volumes of each stock solution (e.g. tryptophan in water and ammonium sulfate in water) with care taken to ensure the solution was well-mixed before nebulization. The composition of the particles is assumed to match that of the non-volatile components of the bulk solution. A full list of the gravimetric solutions used is presented in Table 1.
- A nitrogen tank or HEPA-filtered room air was used to supply pressure to the nebulizer and a sealed dilution volume with inlet and outlet ports. The flows to the nebulizer and the dilution chamber were both controlled with rotameters tuned to provide adequate flow to the nebulizer while sufficiently diluting the output aerosol to maintain a manageable particle sample rate in the WIBS (limited by the recharge time of the flash lamps). The minimum flow rate required for the medical
- nebulizers used here (B&F AeroMist Nebulizer; Allied Healthcare Products, Inc.; St. Louis, MO, USA) was determined to be ~1 liter/min, though this depended slightly on the solution. Typical dilution flow rates were 3-5 liter/min, yielding a total output flow of 4-6 liter/min. A bypassing port (a simple T-union with one end open to ambient) was installed downstream of the dilution chamber and upstream of a diffusion drier containing Drierite (anhydrous CaSO<sub>4</sub>), which reduced the RH
- of the aerosol stream to

The DMA was used to select a narrow size-range of the incoming poly-disperse aerosol for sampling by the WIBS. We calculated the per-particle mass of fluorophore based on the selected particle mobility diameter and the mass fraction of nebulized solution assuming dry spherical particles. This
mass provides the basis for our calibration scheme, as it associates the fluorescence signals from the

WIBS with an absolute fluorescent mass. This experimental setup is shown in Figure 2a.

The flow rate of the aerosol stream through the diffusion drier and the DMA is controlled by the WIBS flowrate (0.3 liter/min) and a mass flow controller (MFC) downstream of the DMA. A sample flow through the DMA of 0.1 liter/min was optimal to select particles in the size range of interest

- (650 nm 3  $\mu$ m). We used a DMA sheath flow of 1 liter/min (10:1 sheath to sample ratio). A make-up flow of 0.2 liter/min of HEPA-filtered lab air was controlled by the MFC after the DMA. It should be noted that this make-up flow was not dried. This 2:1 dilution from the make-up flow did not result in detectable evaporation or uptake of the size-selected particles, as determined by comparing measured particle sizes with and without this dilution, nor did it affect their fluorescence
- signals. However, drying this make-up flow would remove any potential for water uptake, and could potentially be important in more humid environments. An RH probe (INTERCAP HMO60; Vaisala, Helsinki, FIN) installed in-line between the drier and the DMA showed that the measured humidity of the stream was between RH=1-2%, which indicated that particles were thoroughly dried prior to sizing. We estimate the residence time in the drier to be roughly 1.7 minutes.
- In this configuration, the WIBS-4A sampled a stream of mono-disperse aerosol particles with a known mass of fluorescent molecules and the resulting fluorescence signal magnitudes were analyzed. We performed these experiments at several gain voltage settings and report the full results in Section 3. The experiment was conducted within a fume hood to contain all exhaust particles. In the absence of a fume hood, particulate filters on the exhaust lines are recommended to minimize any potential exposure to calibration particles.

#### 2.3 Calibration procedure

The starting point for most calibrations is preparation of the gravimetric fluorescent standard solution. The solutions used in these calibrations are listed in Table 1. We have verified that the signals from quinine and tryptophan do not significantly degrade over the course of two days under the

- preparation and storage procedures used here. We have not assessed the stability of these solutions for longer terms and recommend solution preparation for semi-immediate use. Following solutions preparation, 12-15 mL was transferred to a clean medical nebulizer. This volume of solution was sufficient to last for 1.5 hours, the maximum duration of a typical calibration experiment. Unused solutions were stored in a dark refrigerator.
- Prior to sampling fluorescent particles, the WIBS was run in FT mode for ~5 minutes. Following FT data collection, pressurized nitrogen was provided to the system, starting fluorescent particle generation as described above. The rotameter controlling the dilution flow upstream of the DMA

was adjusted to provide a particle count rate in the WIBS of <100 counts/s, below the maximum duty cycle of the flashlamps (125 counts/s), which allows for detection of all particles. Typical particle

- concentrations in these experiments produed 10-100 counts/s, dependent on solute concentration in the nebulizer, selected mobility diameter, and dilution flow. For each selected mobility diameter, approximately 5 minutes of data were collected, providing enough particles to build high-fidelity histograms for determination of the central value and variability of fluorescence using gaussian fits. After a sufficient amount of data was collected, the voltage on the DMA was changed to select for
- another particle size, and this process was repeated until all data from all target sizes was collected. Data shown in the example experiment depicted in Figure 3 span the particle size range of 650 nm to 2.5  $\mu$ m. Following data collection from size-selected fluorescent particles, FT mode data was collected again for ~5 minutes.

# 2.4 Data analysis

A time series of single-particle data from a typical calibration of the FL2 detector using pure quinine particles is shown in Figure 3a. Black circles show individual fluorescence values and the pink line shows the 30-second average fluorescence intensity for the singly-charged ("Q1") population. Mobility diameters selected by the DMA are labeled with colored bands above the plot. Data are discarded during transitions from one DMA voltage to another (typically requiring ~30 seconds), 215 indicated by small gaps between the colored bands.

We construct fluorescence intensity histograms for each size (shown in Figure 3b) and fit a gaussian function to each singly charged mode, the center of which represents the modal fluorescence intensity for that size. In Figure 3b, each histogram is normalized to its maximum value and colored according to the colored bands in Figure 3a. We calculate the mass of fluorescent material for a

220 given size by assuming a spherical particle shape and complete removal of solvent. In the case of the mixed tryptophan-ammonium sulfate particles, we assume that the mass fraction in the dry particles is the same as in the atomizer solution. Fluorescence intensities as a function of fluorophore mass are then used to construct the calibration curves shown in Figure 4, which are discussed further in the Results section.

#### 225 2.5 Polydisperse fluorescent aerosol calibrations

We also performed fluorescence calibrations using a polydisperse stream of fluorescent aerosol particles. Fluorescent particles were prepared in a similar manner as above with the exception that no DMA was used to size-select from the aerosol stream. This modified experimental set-up is shown in Figure 2b. Typical flowrates are shown in green in Figure 2b, but sometimes much higher dilution flow rates were used to ensure a particle count rate in the WIBS below 125 s<sup>-1</sup>.

Data were analyzed similarly to above, except WIBS scattering signals were used to provide particle size. A smoothed mie curve (so as to be monotonically increasing) based on our instrument's

geometry was used as a sizing calibration for the scattering signals. This sizing calibration curve was derived for each aerosol type used (quinine, ammonium sulfate-tryptophan) and for each gain

- setting based on size-selected data. Each sizing calibration curve was then applied to scattering signals from the polydisperse aerosol data to provide particle diameter, which was converted to mass assuming complete drying and spherical particle shape. It should be noted that the application of a monotonically-increasing size calibration curve necessarily does not capture all of the mie scattering behavior. Thus, due to sizing errors, the polydisperse calibration slopes are expected to be less robust
- than those generated using a DMA. Individual fluorescence signals were binned by size, and gaussian functions were fit to determine the modal fluorescence signal for a given bin. Only data from bins with >500 measured particles are displayed in Figure 4.

# 2.6 Xenon flash lamp intensity tests

- The relationship between fluorescence intensity and excitation power in fluorescence measurements can be complex. For instance, with too-high excitation power there can be saturation effects where fluorescent molecules are photobleached (Faris et al., 1997) or their excited states are depopulated through stimulated emission (Georges et al., 1996). With too-low excitation energy, fluorophores may not exhibit fluorescence at all (Kaye et al., 2005). Thus, the power of the excitation radiation can potentially have a large impact on the magnitude of fluorescent light measured.
- The WIBS-4A provides a measurement of the relative power output of each xenon flash lamp pulse with a fiber optic sensor, which was designed to provide a measure of flash power over the lifetime of the lamp. These sensors, consisting of a silicon PIN-photodiode, which are placed near the arc lamp in the lens tube, provide a current that scales relative to the amount of light measured from each flash. However, they only provide a relative measure of light, and not an absolute measure of the excitation
- energy experienced by each particle detected by the WIBS-4A. Moreover, this measurement is highly sensitive to the placement of the fiber optic sensor within the lens tube of the flash lamp, and may not provide a repeatable measurement when the fiber optic sensor is moved, either through instrument vibrations or when removing the flash lamp module for maintenance. In order to assess the response of our calibration particles to changes in lamp power, we performed fluorescence calibration tests
- using neutral density filters with varying optical densities inserted within the lens tube of the flash lamps. Additionally, we performed our calibrations both with the original flash lamp, which has been installed in our instrument since its purchase ( $\sim$ 3 years ago), and a new flash lamp, in order to assess any possible degradation in power due to prolonged use.

## 3 Results and Discussion

# 265 3.1 Calibration results

Compiled results from the tryptophan and quinine calibrations are shown in Figures 4a and 4b, respectively, constructed as described above from single-particle fluorescence intensities measured for various fluorophore masses. Different symbols correspond to individual calibration experiments and error bars represent the width of the gaussian fits to observed fluorescence intensities for a given

- mass. For both particle types we also tested different PMT detector gain voltages. All gravimetric standard solutions were prepared within two days of a given experiment. We collected FT mode data prior to each experiment, which is also shown on these graphs. The linear fits are constrained such that the y-intercept is equal to the average fluorescence signal from FT mode (second column in Tables 2 and 3.
- The detector gain clearly has a significant impact on the detector response for a given mass of fluorescent material. For instance, for the two FL1 gain settings shown in Figure 4a, the high gain (0.747 V) slope  $(25.0 \pm 1.7 \text{ counts/fg tryptophan})$  is ~4 times higher than for the low gain (0.632 V) slope  $(5.89 \pm 0.32 \text{ counts/fg tryptophan})$ . Thus, the mass of tryptophan that saturates the FL1 detector at the high gain setting is ~4 times lower than the low gain setting. As mentioned earlier,
- two WIBS clustering studies (Robinson et al., 2013; Crawford et al., 2015b) excluded saturating particles from their analysis, and so understanding the range of measurable fluorophore mass is potentially important.

In Figure 4 we show an average linear fit to all data, but the slope of the reported calibration curves in Table 1 is the average slope across all of the individual experiments. Slopes for individual

- calibration experiments, for a given FL channel and gain, were all within 15% of each other, and between 3 and 5 experiments were performed at each gain setting. The stability of these gain curves was assessed over a period of 4 months. For both quinine and mixed tryptophan/ammonium sulfate particles, the relationship between fluorescence intensity and fluorophore mass was linear. We take this as evidence that there is not significant shielding or quenching of fluorescence over the fluo-
- rophore concentrations and particle sizes studied here. While there is noise in the data points shown in Figure 4, there are no systematic biases or individual experiments that are outliers, as the data from all experiments are distributed around the linear fit.

The variability in fluorescence signals in these calibration experiments are represented by error bars in Figure 4. These error bars are the width of the gaussian fits to the fluorescence intensities

at a given particle size. They represent the standard deviation of single-particle fluorescence values observed for a given mobility diameter selected by the DMA and are typically  $\pm$  20%. Because we are atomizing a well-mixed solution, we assume that all particles generated are uniform in composition, and so the variability in fluorescence can be attributed either to inherent noise in the WIBS, the transfer function of the DMA, or some combination of both. The variability in fluorescence inten-