# Peer review of "Fluorescence calibration method for single-particle aerosol fluorescence instruments"

_Atmospheric Measurement Techniques, 2016_

## Referee Comment (RC1) · Anonymous Referee #1 · 10 Nov 2016

With the growth of biological aerosol research in the last decade, commercially-available instruments are now available that utilize fluorescence techniques to quantify bio-aerosol concentrations and properties. This paper provides a framework for calibrating such fluorescence instruments, specifically to provide a method to normalize the fluorescence intensity response over time and between different instruments with different operational settings. This work is well done, well presented, and is a welcomed addition to research in this topic area. I do have some reservations as to the application of normalized fluorescence intensities to ambient data and suggest revising the manuscript to address these hesitations. Otherwise, I support publishing the paper after the authors address the following minor comments:

1. Line-80: Please state which requirements were not fulfilled by NADH and naphtha-lene here. Were any other materials considered and not used? This information might

be useful to other groups seeking additional calibration candidates.

2. Section 2.1: One aspect of WIBS operation not discussed is timing. I suggest adding a short paragraph summarizing how the instrument timing was set, so this procedure can be used consistently by the community.

3. Line-118: It is interesting that peak height varies as a function of flow rate and I certainly agree that calibrations should be done at an appropriate flow. Since the data in Figure 1 seem to asymptote toward a constant response at higher flow rates, would you suggest that users operate at a flow rate > 4 LPM so peak height is less sensitive to small variations in flow rate? Users would have to accept limitations of decreased counting efficiency for high concentrations at these higher flow rates.

4. Line-162: What model DMA was used here, and is it able to size select particles greater than 1 micron diameter?

5. Line-438: Larger, philosophical issue. Expressing fluorescence intensity as T- and Q-units is certainly a clean and easy way of comparing the output of different instruments to each other. But, there is a risk of largely over-simplifying the interpretation of ambient results where many complex factors govern fluorescence intensity. This is noted explicitly in the Pohlker review of bioaerosol autofluorescence, "However, fluorescence intensity is a complex function of various parameters such as concentration, extinction coefficient at $\lambda$ex and quantum yield at $\lambda$em as well as influences by the molecular environment. Accordingly, only semi-quantitative comparison of intensity levels is possible based on the presented results." For example, two ambient populations could result in the same Q-unit fluorescence and have very different actual amounts of fluorescent material because of the properties listed above. Interpreted results that showed similar Q-unit fluorescence intensity would not be at all accurate in this case. By advocating the use of T- and Q-units, are we over-simplifying these systems and risking erroneous interpretation? There is no doubt that using this method to ensure instruments are operating similarly is very beneficial, but I question the ap-

plication to ambient analyses. I suggest adding a caution to users wishing to apply T- and Q-units for ambient applications, potentially including a review of the large range of quantum yields for fluorescent material, and removing this recommendation from the conclusions (Line-453).

Pöhlker, C., J. A. Huffman, and U. Pöschl. "Autofluorescence of atmospheric bioaerosols–fluorescent biomolecules and potential interferences." Atmospheric Measurement Techniques 5.1 (2012): 37-71.

---

## Referee Comment (RC2) · 1 Dec 2016

This journal article presents a method for calibrating the response of fluorescence instruments against a known standard. The work is very timely, with an increase in the availability of commercial instruments and the increased attention biological material is receiving in the research community. The article is well written and describes very clearly the steps required to perform the calibration. I have a few comments below, but otherwise I think this article is well suited for AMT and should be published.

Figure 1. I agree that performing the calibration at the operating flow rate is the way forward, but I think a recommendation of the paper should be that figure 1 is generated/ checked at regular intervals (start and end of campaigns maybe) with the fluorescent material. This would give you an operational baseline to check the instrument perfor-

[Figure]

mance over time. It would also be something that is easily included in supplementary material in publications so different groups can compare sensitivities, if required.

Figure 3a. You have calibration data you are not using. If you know where the Q1 peaks are, you therefore know the location of the Q2 peaks. This is most noticeable at the smaller sizes. You have additional masses from the single mobility diameter. This feature of DMAs is often used when calibrating OPCs with oil drops.

I have read the comments of the other reviewer regarding the Q- and T- equivalent mass. I tend to agree that it potentially over simplifies the measurement, but this approach is used elsewhere in science. For example, the Aerodyne Aerosol Mass Spec community report nitrate equivalent mass, which assumes everything has the same ionisation efficiency as nitrate. If they want the mass of a specific compound, they need to apply a relative ionisation efficiency correction. I think caveating the use of the Q- and T- with other factors that can affect it is required, but it is still a useful quantity to report. Maybe as more research is done, a database of Relative Fluorescent Factors (RFR) will be generated for different materials.

---

## Author Comment (AC1) · 21 Jan 2017

Response to RC1 - amt2016-331

We would like to thank the reviewer for their time looking over our manuscript. This feedback really truly very helpful, both in it being thorough and thoughtful. Thank you very much for reading the paper. We have organized our responses to the reviews by using the same numbering as the initial review, and responding below the reviewer's comments.

***** ***** ***** ***** ***** ***** ***** ***** ***** ***** ***** ***** ***** ***** ***** ***** ***** ***** *****

With the growth of biological aerosol research in the last decade, commercially avail-

able instruments are now available that utilize ïñĆuorescence techniques to quantify bio-aerosol concentrations and properties. This paper provides a framework for calibrating such ïñĆuorescence instruments, speciïñĄcally to provide a method to normalize the ïñĆuorescence intensity response over time and between different instruments with different operational settings. This work is well done, well presented, and is a welcomed addition to research in this topic area. I do have some reservations as to the application of normalized ïñĆuorescence intensities to ambient data and suggest revising the manuscript to address these hesitations. Otherwise, I support publishing the paper after the authors address the following minor comments:

1. Line-80: Please state which requirements were not fulïñĄlled by NADH and naphthalene here. Were any other materials considered and not used? This information might be useful to other groups seeking additional calibration candidates.

-As stated on line 81 "Results from all materials tested are presented in Section 3," so we are not omitting any useful information. A sentence describing the shortcomings of each NADH and naphthalene has been added here to address this concern, while the full details are in section 3.3. This section previously was labeled "Other materials," but we have changed the title to "Failed calibrants: NADH and naphthalene" to set apart these materials from quinine and tryptophan.

2. Section 2.1: One aspect of WIBS operation not discussed is timing. I suggest adding a short paragraph summarizing how the instrument timing was set, so this procedure can be used consistently by the community.

-We have added a short paragraph summarizing how we set the timing. "The timing of the firing of each flash lamp was set using the optimization function in the WIBS acquisition software while sampling monodisperse fluorescent particles, typically FPSLs though the size-selected calibration particles presented here work as well. The timing optimization program scans through a wide range of delay times for the lamps for a given fluorescent channel following triggering (detecting the scattered light pulse). The

software simultaneously averages fluorescent signals. The delay time corresponding to the maximum average fluorescent signal determines the optimal flash lamp timing. Flash lamp timing was periodically determined for each fluorescent channel, but did not vary over the course of these measurements."

3. Line-118: It is interesting that peak height varies as a function of flow rate and I certainly agree that calibrations should be done at an appropriate flow. Since the data in Figure 1 seem to asymptote toward a constant response at higher flow rates, would you suggest that users operate at a flow rate > 4 LPM so peak height is less sensitive to small variations in flow rate? Users would have to accept limitations of decreased counting efficiency for high concentrations at these higher flow rates.

-We appreciate that the peak height for the data in Figure 1 do converge for the highest flow rates presented. However, what we think this really means is that at higher flow rates the instrument's ability to resolve the true scattering peak height is getting worse and worse. So, while at low flow rates we are more sensitive to fluctuations in flow, the instrument is better able to resolve peak heights. Luckily, we don't expect large fluctuations in flow through the instrument in most sampling applications, as the sample and sheath flowrates are controlled by precision flow controllers with stated accuracies within 1% of the reading. So we don't expect this issue to be a real concern, and in fact have reason to operate at lower flow rates when possible. Our goal in presenting Figure 1 was to illustrate an issue that exists for the WIBS-4A model, and emphasize that fluorescence and size calibrations are only valid for a given flow configuration. The limitations of a given sampling situation may necessitate operating the WIBS at different flow rates, which would require different calibrations.

4. Line-162: What model DMA was used here, and is it able to size select particles greater than 1 micron diameter?

-As stated in the original text, the DMA used here is custom-built, and yes it can select particles greater than 1 um. There is no previous instrument paper to cite for the

[Figure]

NOAA DMA, though we have added to the text that the NOAA DMA column is longer than e.g. TSI 3081, which gives it the ability to select larger particles. See P6 L171 of the updated text for a mention of this: "It should be noted that the custom-built DMA has a longer column than some commercially-available versions (e.g. TSI 3081), and so is more easily able to select larger size particles."

5. Line-438: Larger, philosophical issue. Expressing fluorescence intensity as T- and Q-units is certainly a clean and easy way of comparing the output of different instruments to each other. But, there is a risk of largely over-simplifying the interpretation of ambient results where many complex factors govern fluorescence intensity. This is noted explicitly in the Pohlker review of bioaerosol autofluorescence, "However, fluorescence intensity is a complex function of various parameters such as concentration, extinction coefficient at $\lambda$ex and quantum yield at $\lambda$em as well as influences by the molecular environment. Accordingly, only semi-quantitative comparison of intensity levels is possible based on the presented results." For example, two ambient populations could result in the same Q-unit fluorescence and have very different actual amounts of fluorescent material because of the properties listed above. Interpreted results that showed similar Q-unit fluorescence intensity would not be at all accurate in this case. By advocating the use of T- and Q-units, are we over-simplifying these systems and risking erroneous interpretation? There is no doubt that using this method to ensure instruments are operating similarly is very beneficial, but I question the application to ambient analyses. I suggest adding a caution to users wishing to apply T- and Q-units for ambient applications, potentially including a review of the large range of quantum yields for fluorescent material, and removing this recommendation from the conclusions (Line-453).

-We completely agree with the reviewer on most of the points made in this comment. For example, "two ambient populations could result in the same Q-unit fluorescence and have very different actual amounts of fluorescent material." This is absolutely correct, for the reasons cited above in Pohlker, et al. However, we do not quite understand

sentence that follows, "Interpreted results that showed similar Q-unit fluorescence intensity would not be at all accurate in this case." What is being referred to that is not accurate? The ultimate goal of this calibration method is not to quantify the amount of fluorophore mass in e.g. ambient particles by measuring their fluorescence with the WIBS. Rather, the goal in using this calibration method is to create a scale that is not completely arbitrary, and thus can be used over time and across instruments. Using "Q-units," or something similar, aerosol populations from different datasets could be compared to each other, and it would be possible to say "population A in study 1 has the same fluorescence as population B in study 2." That is the step forward we hope that this paper makes possible. It would be an erroneous leap to go one step further, as the reviewer points out, to say that the amount of fluorescent mass in population A and population B are the same, but that is not what our paper is advocating for. The last paragraph of section 3.5 makes this clear, where we relate the fluorescence from Blue 1 micron FPSLs to the fluorescence of a mass of quinine under the operating conditions of our instrument. But we are not saying we have any knowledge of the mass of the actual fluorophores in the PSL particles.

Pöhlker, C., J. A. Huffman, and U. Pöschl. "Autofluorescence of atmospheric bioaerosols–fluorescent biomolecules and potential interferences." Atmospheric Measurement Techniques 5.1 (2012): 37-71.

---

## Author Comment (AC2) · 21 Jan 2017

Response to RC2 - amt2016-331

We would like to thank the reviewer for their time looking over our manuscript. This feedback really truly very helpful, both in it being thorough and thoughtful. Thank you very much for reading the paper. We have organized our responses to the reviews by using the same numbering as the initial review

***** ***** ***** ***** ***** ***** ***** ***** ***** ***** ***** ***** ***** ***** ***** ***** ***** ***** *****

This journal article presents a method for calibrating the response of fluorescence instruments against a known standard. The work is very timely, with an increase in the

[Figure]

availability of commercial instruments and the increased attention biological material is receiving in the research community. The article is well written and describes very clearly the steps required to perform the calibration. I have a few comments below, but otherwise I think this article is well suited for AMT and should be published.

Figure 1. I agree that performing the calibration at the operating flow rate is the way forward, but I think a recommendation of the paper should be that figure 1 is generated/ checked at regular intervals (start and end of campaigns maybe) with the fluorescent material. This would give you an operational baseline to check the instrument performance over time. It would also be something that is easily included in supplementary material in publications so different groups can compare sensitivities, if required.

-We agree, and hope that the size calibration and, with this new method, fluorescence calibration, are checked and reported for measurements for the purposes of verifying an instrument's operational baseline, and allowing other users to better interpret their results.

Figure 3a. You have calibration data you are not using. If you know where the Q1 peaks are, you therefore know the location of the Q2 peaks. This is most noticeable at the smaller sizes. You have additional masses from the single mobility diameter. This feature of DMAs is often used when calibrating OPCs with oil drops.

-It is true that the doubly-charged particles can provide additional data points for this calibration and others like it. We, however, did not optimize our sampling to make the Q2 data useful. In short, we didn't sample long enough (collect enough particles) to make high-fidelity histograms that we could then fit well with Gaussian functions for all of the particle sizes used here. In our data analysis, we found it easier to simply focus on Q1 peaks instead of sometimes also using the Q2 data. This is the kind of improvement on the method we present here as a template that other groups may wish to incorporate in their adaption of it, should they choose to.

I have read the comments of the other reviewer regarding the Q- and T- equivalent mass. I tend to agree that it potentially over simplifies the measurement, but this approach is used elsewhere in science. For example, the Aerodyne Aerosol Mass Spec community report nitrate equivalent mass, which assumes everything has the same ionisation efficiency as nitrate. If they want the mass of a specific compound, they need to apply a relative ionisation efficiency correction. I think caveating the use of the Q- and T- with other factors that can affect it is required, but it is still a useful quantity to report. Maybe as more research is done, a database of Relative Fluorescent Factors (RFR) will be generated for different materials.

-The analogue of 'nitrate-equivalent mass' within the AMS community is roughly what we had in mind in presenting these 'Q-units.' We completely agree with the other reviewer that the intensity of measured fluorescent light is complex and governed by many factors, environmentally-dependent quenching being one such example. We still feel that Q-units, or something similar, is a step forward because it allows comparing measures of fluorescence across days (in ambient sampling) and across different instruments (in ambient or lab sampling). That is not currently possible with the arbitrary fluorescence units usually reported in WIBS studies. At least with Q-units, with all of the necessary caveats clearly stated, measurements can be compared across platforms.